# Bacterial diversity of cantaloupes and soil from Arizona and California commercial fields at the point of harvest

Madison Goforth[1,2], Victoria Obergh[1,2], Richard Park[1,2], Martin Porchas[2,3], Paul Brierley[2,3], Tom Turni[2,4], Bhimanagouda Patil[2,5], Sadhana Ravishankar[1,2], Steven Huynh[6], Craig T. Parker[6], Kerry K. Cooper[1,2,7] *

1 School of Animal and Comparative Biomedical Sciences, University of Arizona, Tucson, Arizona, United States of America, 2 University of California, Agricultural and Natural Resources, Cooperative Extension, Fresno, California, United States of America, 3 Vegetable and Fruit Improvement Center, Department of Horticultural Sciences, Texas A&M University, College Station, Texas, United States of America, 4 Produce Safety and Microbiology, Agricultural Research Services, USDA, Albany, California, United States of America, 5 USDA, Center of Excellence, Melons, Vegetable and Fruit Improvement Center, Texas A&M University, College Station, Texas, United States of America, 6 Yuma Center of Excellence for Desert Agriculture, University of Arizona, Yuma, Arizona, United States of America, 7 BIO5 Institute, University of Arizona, Tucson, Arizona, United States of America

* kcooper@arizona.edu

**Data Availability Statement:** All sequenced reads generated during this study have been deposited in NCBI's SRA archive under the Accession number:

## Abstract

Across the United States, melons are a high demand crop reaching a net production of 2.7 million tons in 2020 with an economic value of $915 million dollars. The goal of this study was to characterize the bacterial diversity of cantaloupe rinds and soil from commercial melon fields at the point of harvest from two major production regions, Arizona, and California. Cantaloupes and composite soil samples were collected from three different commercial production fields, including Imperial Valley, CA, Central Valley, CA, and Yuma Valley, AZ, at the point of harvest over a three-month period, and 16S rRNA gene amplicon sequencing was used to assess bacterial diversity and community structure. The Shannon Diversity Index showed higher diversity among soil compared to the cantaloupe rind regardless of the sampling location. Regional diversity of soil differed significantly, whereas there was no difference in diversity on cantaloupe surfaces. Bray-Curtis Principal Coordinate Analysis (PCoA) dissimilarity distance matrix found the samples clustered by soil and melon individually, and then clustered tighter by region for the soil samples compared to the cantaloupe samples. Taxonomic analysis found total families among the regions to be 52 for the soil samples and 12 among cantaloupes from all three locations, but composition and abundance did vary between the three locations. Core microbiome analysis identified two taxa shared among soil and cantaloupe which were *Bacillaceae* and *Micrococcaceae*. This study lays the foundation for characterizing the cantaloupe microbiome at the point of harvest that provides the cantaloupe industry with those bacterial families that are potentially present entering post-harvest processing, which could assist in improving cantaloupe safety, shelf-life, cantaloupe quality and other critical aspects of cantaloupe post-harvest practices.

SRR25168139 and are associated with BioProject Accession number: PRJNA957757.

**Funding:** This study was supported by a United States Department of Agriculture (USDA), National Institute of Food and Agriculture (NIFA), Specialty Crop Research Initiative (SCRI) award: USDA-NIFA-SCRI #2017-51181-26834 award to the National Center of Excellence for Melon at the Vegetable and Fruit Improvement Center of Texas A&M University including but not limited to PI: Bhimanagouda Patil, co-PIs: Kerry Cooper, Paul Brierley, Tom Turini, and Sadhana Ravishankar. It was also supported by the Technology and Research Initiative Fund (TRIF) provided to Kerry Cooper by the University of Arizona. No funding agency had any role in the study design, data collection and analysis, decision to publish, or preparation of the manuscript.

**Competing interests:** The authors have declared that no competing interests exist.

## Introduction

Within the United States, melons are a multi-million-dollar commodity crop, generating $915 million dollars in 2020 from the production of 2.7 million tons of melons [1, 2]. Cantaloupe and watermelon are the main varieties produced and consumed within the United States, with watermelon production being the larger of the two with 1.6 million tons of domestic production, while cantaloupe had 508,885 tons [1]. Cantaloupe production in the US is predominantly conducted in two states, California, and Arizona, providing 438,000 tons or approximately 90% of domestic production, while several additional states make up the remaining 10% [1]. The remaining 0.7 million tons of US melon production are composed of miscellaneous or specialty melons [1].

Melons have continued to be a popular commodity crop within the United States, where per capita annual consumption is approximately 21 pounds [1]. Melons are a sugary treat that are high in beta-carotene, and have been bred to have these characteristics for consumer consumption [2, 3]. Most varieties of melons are low-maintenance and are suited for warmer climates. However, there are two main categories, summer and winter melons that allow for the commodity to be produced year-round [4–6]. However, the North American cantaloupe variety has been adapted to California and Arizona climates to have two growing periods as a summer melon. There are two growing seasons for each state. California starts in April and goes on until August, with the second season going from October to December [7]. Arizona starts in January and goes on until May, with the second season going from July to November [8]. Knowing the temporal geography of where 90% of the U.S. cantaloupes are grown allows for less variation in regional growth when characterizing the bacterial communities on these surfaces.

To date, there have been a limited number of studies examining the microbiome of the melon flesh, rind, or stem at the point of harvest, utilizing non-culture methodologies like 16s rRNA gene amplicon or shotgun metagenomics. Saminathan et al sampled mature watermelon flesh from six cultivars grown at West Virginia State University, and found a high relative abundance of both *Enterobacteriaceae* and *Pseudomonadaceae* with differences between these bacterial families among the different melon cultivars [9]. Franco-Frias et al sampled cantaloupe rind rinsates, worker hand rinsates, and soil samples from two farms in Coahuila, Mexico. The authors found with Principal Coordinate analysis (PCoA) grouping of cantaloupe and hand rinsates samples by farm compared to soil samples that clustered by visit rather than the farm [10]. Xiao et al examined the rhizospheres, bulk field soil, and stems of oriental and netted melons grown in the same location in China. The authors found that the rhizospheres of the netted and oriental melons, and bulk soil samples did not have any statistically significant alpha diversity differences, but the PCoA analysis resulted in clustering of the samples by type rather than any other factor [11]. All the studies mentioned above are in relation to melons, but do not provide an aspect of the melon microbiome at the point of harvest from commercial fields and the variation between different growing locations.

There are about 48 million cases each year of foodborne illness in the US, which is equivalent to one in six Americans becoming sick [12]. Since 1971, there have been 1,746 outbreaks related to melons, with 35,097 illnesses, 4,287 hospitalizations, and 74 deaths [3, 13]. While there have been major strides in improving melon safety for consumers, there are still food safety issues as demonstrated by the 2011 *Listeria monocytogenes* outbreak that involved 147 cases, 143 hospitalizations, and 33 deaths from the cantaloupe industry in Colorado [14–17]. Potential contamination routes from this multistate outbreak include the transportation trucks, a used potato-washing machine, and a pool of water near the packaging equipment in the storage facility [18, 19], which are all post-harvest contamination risks. Given these risks, it

is still important to understand the cantaloupe microbiome at the point of harvest to be aware of those microbial communities that are entering the post-harvest process on the melon rind and how it varies in different states/locations. This awareness of the melon microbiome entering post-harvest will help to develop more effective ways to reduce these risks of post-harvest pathogen contamination during various stages of processing across different states/locations, especially since a single outbreak can devastate production levels and economic gain in other states besides where the contaminated melons were grown.

Overall, the critical control points for melons post-harvest typically consist of packaging and persistent cooling in between transportation and shipments to wholesale and retail stores for consumption [16, 19, 20]. Contamination control methods for melons can include being sanitized by various detergents or solutions that include hydrogen peroxide and keeping melons on a flotation device until shipped for the final destination [16, 19]. There are different routes to take depending on the farm and retail service requesting the melons, which means the number of and types of post-harvest steps like washing and cooling between the farm and the consumer changes. These variations in steps can also increase the risk for foodborne pathogen contamination and survival during transportation [14, 19]. Other factors that can contribute to pathogen contamination of melons include wash water quality, melon surface moisture, and sanitation of packaging handlers and facilities [19, 21, 22]. Understanding those microbial communities that are present on the melon surface at the point of entering post-harvest processing establishes a foundation for potential pathogen-microbe interactions that can occur during the post-harvest processing steps at higher risks of contamination. These microbial communities play a role in potentially encouraging or preventing pathogen contamination, but without a strong understanding of the general composition of melon microbiome, how it varies in different locations, and the influence that environmental factors (e.g. soil, water, etc.) may have on the microbiome, there is a critical food safety knowledge gap that also limits studying pathogen-microbe interactions that could occur during the post-harvest processing. Additionally, since the Hazard Analysis Critical Control Point (HACCP) protocols for U.S. grown melons vary, characterizing the bacteria on the surface at the point of harvest in different locations can help create a more standardized protocol that could reduce the risk of potential pathogen contamination.

The overall goal of this study was to investigate the bacterial diversity on the rinds of cantaloupes and field soil from commercial cantaloupe farms in three different major growing regions of the US. Specifically, our study focused on the following: 1) characterizing the bacterial diversity and composition of the netted rind of commercial cantaloupes; 2) characterizing the bacterial diversity and composition of the commercial cantaloupe field soil; 3) evaluating the regional differences of Arizona and California bacterial diversity between the different samples; 4) and inferring potential similarities and differences in bacterial composition between soil and cantaloupe samples.

## Materials and methods

### Sample collection and DNA extraction

Melons and soil samples were collected in 2019 directly from commercial cantaloupe fields in three locations during summer growing regions: (1) Central Valley, California; (2) Imperial Valley, California; (3) Yuma Valley, Arizona. Samples were packed on ice and transported back to the University of Arizona for further processing as described below. The total number of each type of sample collected is listed in Table 1. Composite soil samples with a weight of approximately 25 grams were collected in a sterile WhirlPak® bag from five sites (approximately five grams per site) within one meter of the location that cantaloupes were also

Table 1. Soil and cantaloupe samples collected during the study period.

| | Growing Region | | | |
|---|---|---|---|---|
| | Central Valley | Imperial Valley | Yuma Valley | Total |
| Melons | 29 | 10 | 30 | 69 |
| Soil | 10 | 14 | 43 | 67 |
| Total | 39 | 24 | 73 | 136 |

sampled, and then gently hand massaging the soil for 3 minutes to mix the soil. Each cantaloupe was sampled by a sterile swab (BD Diagnostics) that was dipped in a sterile detergent solution (0.15 M NaCl, 0.1% Tween-20) (Fisher Scientific, Pittsburgh, Pennsylvania) [23], which helps to remove the bacteria from the surface of the cantaloupe rind, and then the entire surface of the cantaloupe was swabbed except for surface in direct contact with the soil. Cantaloupes were individually sampled within a field during the sampling time point, and cantaloupes were not moved from their field growing location at any point during swab sampling. DNA was extracted from each composite soil sample and cantaloupe swab using the Qiagen DNeasy PowerSoil Pro kit (Qiagen, Hilden, Germany) per the manufacturer's instructions with modifications; (1) samples were incubated at 65°C for ten minutes after the addition of C1; (2) and an elution of the DNA from the column filter with 50 µL of C6 added twice with a centrifugation step in between to increase yield from the manufacturer's instructions.

## 16S rRNA PCR amplification and Illumina library preparation

The V4-V5 region of the 16S rRNA gene was PCR amplified using the barcoded 515F-926R primers in triplicate 25 µL reactions. Each 25 µL reaction consisted of 10 µL of 2x Platinum Hot Start DNA Polymerase (Thermo Fisher Scientific, Waltham, MA), 1 µL of premixed forward and reverse primers (10 µM) (IDT, Coralville, IA), 5 µL of template DNA, 6.5 µL of PCR grade nuclease-free water (Qiagen # 17000–10), 1.25 µL of mPNA blocker (5 µM; PNA Bio, mitochondria blockers), and 1.25 µL of pPNA blocker (5 µM; PNA Bio, chloroplast blockers). Nuclease-free water was used as a negative control for PCR amplification (Thermo Fisher Scientific, Waltham, MA). The PCR reaction consisted of 95°C for 3 minutes, followed by 30 cycles of 45s at 95°C, 45s at 50°C for each cycle, 90s at 68°C, and a final amplification step of 68°C for 5 minutes. After amplification, triplicate samples were pooled together and visualized on a 1.5% agarose gel to confirm proper amplification of each sample and no contamination in negative controls. Each amplified sample was quantified using the Quant-iT PicoGreen® assay kit (Thermo Fisher Scientific, Waltham, MA) per the manufacturer's instructions, and then all samples pooled together in equal molar ratios. Pooled barcoded sequence libraries were then cleaned using the QIAquick PCR cleanup kit (Qiagen, Hilden, Germany) per manufacturer's protocol, and then shipped to the Produce Safety and Microbiology Section, Agricultural Research Service, United States Department of Agriculture for sequencing on an Illumina MiSeq instrument. Barcoded libraries were sequenced using the Illumina MiSeq reagent v3 (600 cycle) kit to generate 300 bp paired end reads. All sequenced reads generated during this study have been deposited in NCBI's SRA archive under the Accession number: SRR25168139 and are associated with BioProject Accession number: PRJNA957757.

## Sequencing read processing

Sequence reads were demultiplexed, quality trimmed, and denoised in QIIME2 (v2020.2) [24] software with the DADA2 plugin [25]. Sequences were trimmed at the point that the quality score dropped below Q30 for forward reads (300 bp) and reverse reads (201 bp), and then

paired reads were merged together to generate a single read with a 90 bp overlap. Merged reads were then used for further analysis in QIIME2 including phylogenetic analysis and taxonomic analysis as described below.

## Taxonomic assignment

Taxonomic analysis was conducted using the QIIME2 software feature-classifier commands and Greengenes database (v13.8) to classify the merged reads with 99% sequence similarity. The classifier was trained to the 515F – 926R primers with a minimum base pair length of 200 and maximum base pair length of 500 and using the following sequences for each primer: 515F primer 5'-GTGYCAGCMGCCGCGGTAA-3' and 926R primer 5'-CCGYCAATTYMTTTRAGTTT-3'.

## Alpha and beta diversity analysis

The QIIME2 generated amplicon sequence variant (ASV) file and phylogenetic tree file were then exported for further analysis in R software (v4.2.1) along with the metadata file for all samples. Analysis in R included the usage of the following packages: (1) phyloseq (v.1.40.0) [26], (2) microbiome (v.1.18.0) [27], and (3) vegan (v.2.6.2) [28]. Imported files were used to first generate a phyloseq object using the phyloseq package, which was then filtered for any chloroplasts and mitochondria sequences that were not blocked during amplification. Sample types like soil and melons were divided and rarefied individually for site comparison. All soil samples were rarefied at 20,000 reads with 10,155 taxa retained and 14 of the 67 total samples dismissed. All melon samples were rarefied at 1,500 reads (due to low biomass of collecting individual melon samples as described above, instead of composite samples) with 1,820 taxa retained and 21 from the 69 total samples dismissed. When comparing soil to melon samples, each sample was rarefied to 1,500 reads with 8,102 taxa left and 21 of the 136 total samples dismissed (S1 Table in S2 File). Alpha and beta diversity analysis was conducted using phyloseq, microbiome, and microbiomeutilities R packages [26, 29] including the Shannon Diversity Index, Bray-Curtis dissimilarity distance matrix, and data visualization. Other alpha diversity matrices were also conducted that included Chao1 total richness, Simpson evenness, and Simpson dominance index. Statistical inference for alpha diversity used non-parametric Kruskal-Wallis pairwise test with a post-hoc Wilcox test using phyloseq and stats package. The adonis2 function in the vegan R package with 999 permutations was used to calculate the statistical variability in bacterial composition among the samples.

## Taxonomic composition visualization, and core microbiome analysis

Taxonomic composition visualization was generated using the phyloseq, microbiome, microbiomeutilities, and vegan packages in R. Parameters were set to 0.1% detection and 75% prevalence for taxonomic composition of different bacterial families for soil and cantaloupe samples separately, regional variation, and overall core shared among all samples. Core microbiome analysis identified bacterial families that were shared between at least 75% of all the samples used in the analysis, and were identified using the parameters of 0.1% detection and 75% prevalence with the phyloseq R package. The core microbiome was determined for soil and cantaloupe samples separately, regional variation, and overall core shared among all samples. Due to a lack of core microbiome in the soil samples, an extremely relaxed detection parameter of $1 \times 10^{-9}$ was used instead of 0.1% detection for the identification of any core bacterial families in soil samples.

### Additional data analysis

For determining top taxonomic features to assess effect size, the features were analyzed using a Linear Discriminant Analysis (LDA) Effect Size (LEfSe). This analysis used the lefser (v.1.16.0) [30] R package in addition with microbiomeMarker (v.1.3.2) [30, 31] to calculate the effect size of soil and cantaloupe samples with an LDA cutoff of 4 [32]. LEfSe analysis was conducted to determine the taxonomic features that accounted for the differences in only the soil communities, only the melon communities, and then both soil and melon communities based on the different growing regions (Central Valley, Imperial Valley, and Yuma Valley) sampled in this study.

## Results

### Soil microbiome

The Shannon Diversity Index (alpha diversity) showed that soil from Yuma Valley had the highest diversity among the three growing regions, and there was a statistically significant difference between Yuma and Imperial Valleys compared to Central Valley (Kruskal-Wallis P-value <0.01) but not Imperial Valley compared to Yuma Valley (Kruskal-Wallis P-value >0.01; Fig 1A). Alpha matrices like Chao and Simpson further supported the Shannon Diversity Index, indicating that Yuma and Imperial Valley had a higher bacterial diversity, and Central Valley had low bacterial diversity (S1A–S1D Fig). Furthermore, the principal coordinate analysis based on the Bray-Curtis dissimilarity index (beta-diversity) further demonstrated the soil samples clustered by field location ($R^2$ value: 0.25; Permanova P-value 0.001; Permutations

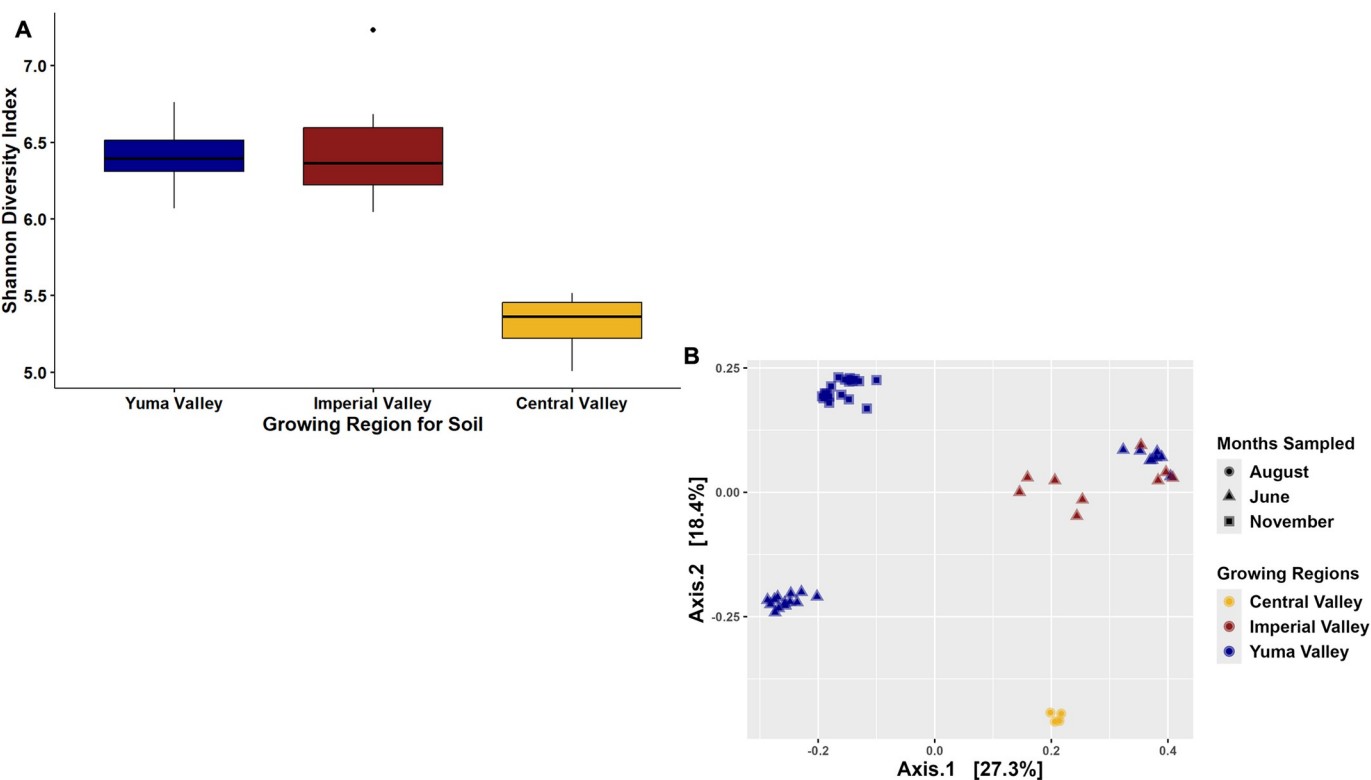

**Fig 1. Bacterial diversity of soil collected in two commercial agricultural states.** (A) Shannon Diversity Index plotted based on the site for soil samples. (B) Bray-Curtis PCoA plot clustered by the region where the soil was collected.

999; Fig 1B), although there was some clustering of the soil between the Imperial and Yuma Valley samples, further supporting the alpha diversity results. Interestingly, there were soil samples collected at two different time points (June and November 2019) for the Yuma Valley that did not cluster completely together, which demonstrates there could be temporal influences on the bacterial communities in commercial cantaloupe field soil ($R^2$ value: 0.239; Permanova P-value 0.001; Permutations 999; Fig 1B).

Taxonomic composition of the soil from the three sites yielded 52 different bacterial families (75% prevalence, 0.1% abundance), but was reduced to 13 bacterial families when the using a higher and more selective 1% abundance. The top four bacterial families present in all three growing regions based on relative abundance were *Bacillaceae*, *Rhodospirillaceae*, *Streptomycetaceae*, and *Cytophagaceae*. The top three bacterial families for both Yuma and Imperial Valleys based on relative abundance were *Bacillaceae*, *Chitinophagaceae*, and *Gaiellaceae*. This taxonomic compositional analysis further supported the differences between Yuma and Imperial Valleys and Central Valley soil samples, as the top three bacterial families based on relative abundance were *Micrococcaceae*, *Planococcaceae*, and *Rhodospirillaceae* (Fig 2A). Furthermore, Central Valley had *Micrococcaceae* and *Planococcaceae* at higher relative abundances compared to the other two valleys that probably contributed to the lower overall bacterial diversity in these soil samples. Interestingly, LEfSe analysis identified more taxonomic features that were responsible for differences in the Central Valley samples with ten features compared to five for Imperial Valley and seven for Yuma Valley. The top two taxonomic features for Central Valley were *Micrococcaceae* and *Planococcaceae*, while the top two for Imperial Valley

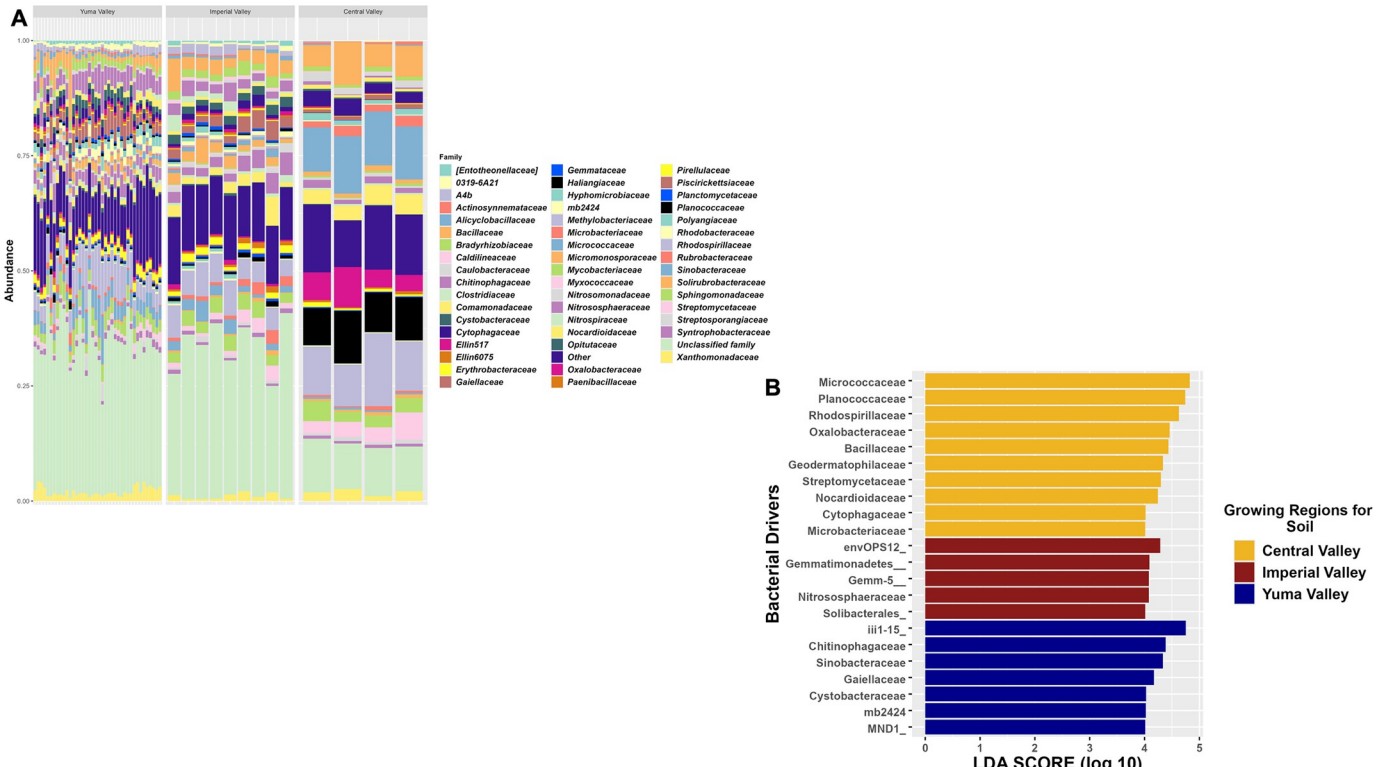

**Fig 2. Bacterial composition of soil collected in two commercial agricultural states.** (A) Taxonomic relative abundance looking at the regions for soil samples, including the top 51 bacterial families in the samples. The "Other" category is composed of 131 bacterial families. (B) Lefser analysis for soil samples looking at sites (LDA = 4).

were order:envOPS12_family:other and order:*Gemmatimonadetes*_family:other and for Yuma Valley were order:*iii1-15* and *Chitinophagaceae* (Fig 2B). These taxonomic results were further supported by core microbiome analysis, which identified 22 bacterial families shared among all regions that were shown in the relative abundance plot except for *JG30-KF-CM45* and *iii1-15* (S2 Table in S2 File). These two missing bacterial families were identified at Order level, and thus were grouped in "Unclassified Family" in the taxonomic plot. For the individual regions, 29, 47, and 43 core bacterial families were found for Yuma, Imperial, and Central Valley Growing Regions, respectively (S3-S5 Tables in S2 File). Overall, *Chitinophagaceae* was more prominent in Yuma Valley and Imperial Valley compared to Central Valley, but *Bradyrhizobiaceae* was consistent among all sites for soil.

## Cantaloupe microbiome

Similar to the soil samples, the Shannon Diversity Index for the cantaloupes from the three different regions found that cantaloupes from Yuma Valley had the highest bacterial diversity. However, there was only a small significant difference for cantaloupes from Yuma Valley versus cantaloupes from Central Valley (Kruskal-Wallis P-value < 0.05; Fig 3A), whereas Yuma Valley or Central Valley versus Imperial Valley showed no significant differences in bacterial diversity. Chao and Simpson richness and evenness matrices further supported the Shannon Diversity Index, indicating that there were little differences in bacterial diversity among the regions for melon bacterial diversity (S2A–S2D Fig). In addition, Chao total richness supports total taxa, including event counts as zero, however, melons overall had around ten times less ASVs compared to soil (S1B & S2B Figs). This was further supported by the principal

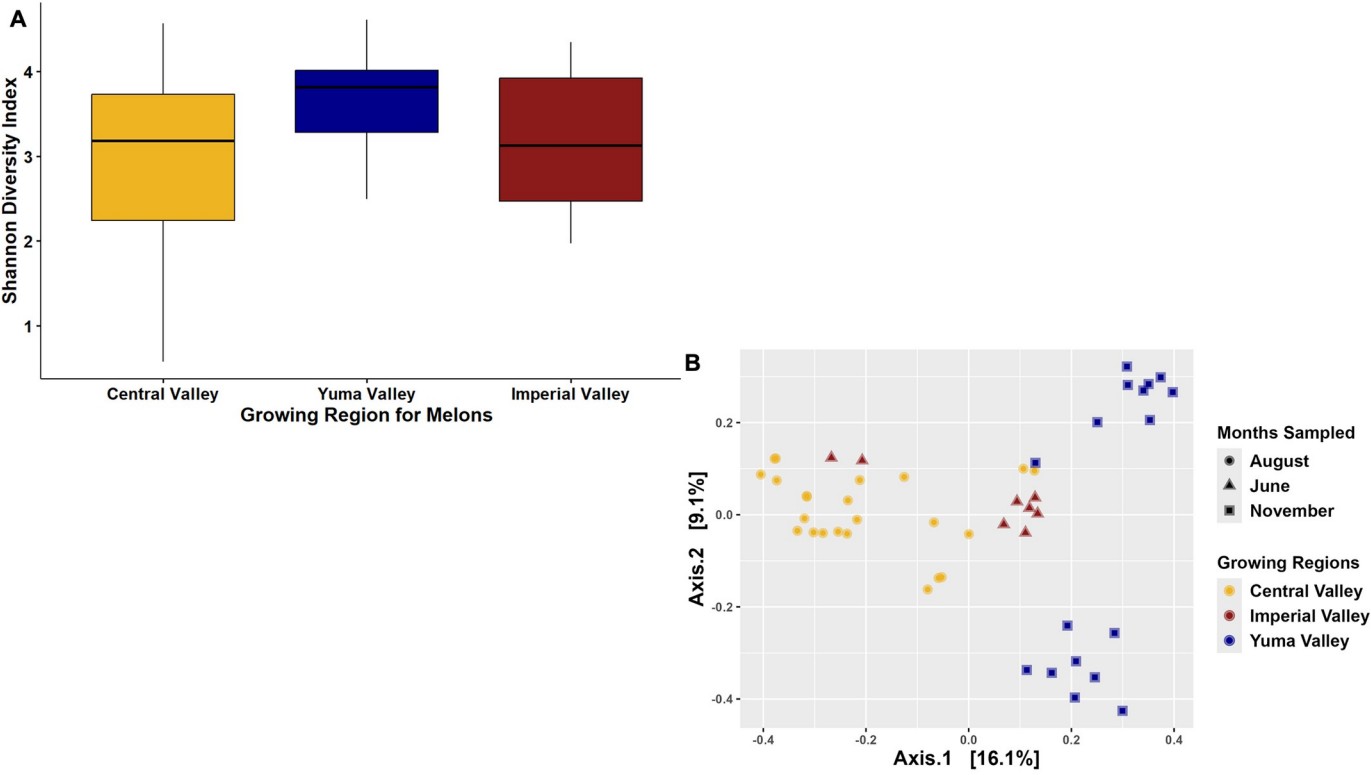

**Fig 3. Bacterial diversity of cantaloupe collected in two commercial agricultural states.** (A) Shannon Diversity Index plotted based on the site for cantaloupe samples. (B) Bray-Curtis PCoA plot clustered by the region where the cantaloupe was collected.

coordinate analysis based on the Bray-Curtis dissimilarity matrix, where the samples did not cluster tightly based on growing region. Although there was clustering between the Imperial Valley and Central Valley cantaloupes, Yuma Valley cantaloupes did cluster away from the other two regions although they were split into two different clusters ($R^2$ value: 0.37; Permanova p-value 0.001; Permutations 999; Fig 3B). The results are interesting considering that Imperial Valley and Yuma Valley are physically closer than Central Valley, suggesting that cantaloupes from those locations should cluster together. It does suggest there is more than environmental/climate factors that could have an influence on the microbiome of cantaloupes in commercial fields.

Taxonomic composition of the cantaloupes for the three sites yielded 12 bacterial families shared among the regions with the top three bacterial families based on relative abundance being *Bacillaceae*, *Enterobacteriaceae*, and *Streptomycetaceae*. The taxonomic analysis further supported the alpha diversity analysis that Yuma Valley had the highest level of diversity, and also supported the beta diversity analysis in that some cantaloupes from Imperial Valley appeared to share some taxonomic profile similarities with Central Valley while others more with Yuma Valley. Overall, the top three bacterial families for Yuma Valley based on relative abundance were *Microbacteriaceae*, *Micrococcaceae*, and *Nocardioidaceae*, those of the Imperial Valley were similar to the top relatively abundant bacterial families, and those in the Central Valley were *Bacillaceae*, *Enterobacteriaceae*, and *Rhodospirillaceae* (Fig 4A). Similar bacterial families were found on all melon samples that were also present in soil except one family, *Enterobacteriaceae*. This bacterial family was found in all regions, but only on

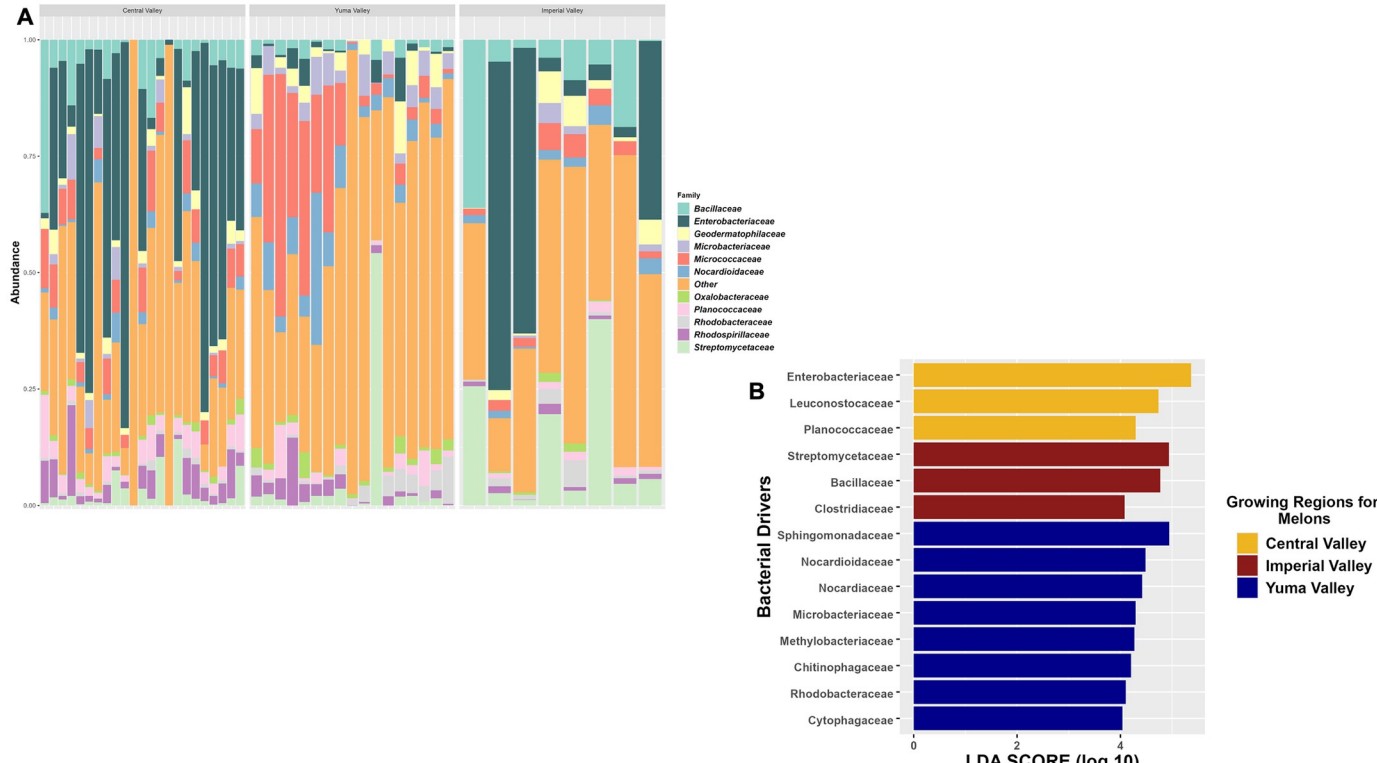

**Fig 4. Bacterial composition of cantaloupe collected in two commercial agricultural states.** (A) Taxonomic relative abundance looking at the regions for cantaloupe samples, including the top 11 bacterial families in the samples. The "Other" category is composed of 151 bacterial families. (B) Lefser analysis for cantaloupe samples looking at sites (LDA = 4).

cantaloupe surfaces and not in soil, in fact even at extremely low relative abundance levels (1 x$10^{-9}$) it was not found in soil samples, which indicated it was well below detection limits. LefSe analysis of the cantaloupes identified more taxonomic features for Yuma Valley at eight features compared to three for Central Valley and three for Imperial Valley. *Enterobacteriaceae* was the main feature for Central Valley, but prominent in all regions when sub-setting the samples to region specific. While Yuma Valley cantaloupes had more taxonomic features compared to the other regions, it only had a single feature, *Chitinophagaceae*, also identified in the soil collected in that region. Central Valley shared one feature between the two sample types, which was *Planococcaceae*, and the Imperial Valley samples did not share any features (Fig 4B). Core analysis at detection of 0.1% identified only a single bacterial family, *Micrococcaceae* (*Arthrobacter*) (S6 Table in S2 File), shared among cantaloupes from all three regions. For individual regions, Central Valley had the most with four core bacterial families (S7 Table in S2 File), followed by both Imperial and Yuma Valley having three core bacterial families (S8 and S9 Tables in S2 File).

### Among all growing regions

As cantaloupes are grown directly on the soil, we decided to investigate the potential bacterial communities that could be transferred from soil onto the melon surface by comparing the two sample types and the taxonomic features that are shared between them. All soil samples were collected near the cantaloupe samples at the same time period for each of the three regions, therefore providing this opportunity to see those members of the cantaloupe microbiome that could potentially be coming from the soil or vice versa. Shannon Diversity Index between both sample types found soil having a greater bacterial diversity compared to cantaloupes overall. Statistical variance found a greater difference between sample type rather than between regions, although it was still significant (Kruskal-Wallis P-value: 2.2x$10^{-16}$ and P-value: 2.50x$10^{-8}$, respectively; Fig 5A). Wilcox post-adjusted confirmed significant differences among the soil and cantaloupes samples for each region and within each region, but not every region had significant variances among the melon samples between regions. Central Valley cantaloupes were significantly different from Central Valley soil (p-value:3.2x$10^{-7}$), Imperial Valley soil (p-value: 4.9x$10^{-9}$), and Yuma Valley soil (p-value: $< 2.16$x$10^{-16}$). Imperial Valley cantaloupes were significantly different from Central Valley soil (p-value: 6.9x$10^{-4}$), Imperial Valley Soil (p-value: 9.4x$10^{-5}$), and Yuma Valley soil (p-value: 4.7x$10^{-8}$). Yuma Valley cantaloupes were also significantly different from Central Valley soil (p-value: 3.6x$10^{-6}$), Imperial Valley soil (p-value: 1.1x$10^{-7}$), and Yuma Valley soil (p-value: 7.7x$10^{-14}$). Central Valley soil was also significantly different from Imperial Valley soil (p-value: 1.5x$10^{-5}$) and Yuma Valley soil (p-value: 1.5x$10^{-9}$). Simpson and Chao alpha diversity metrices supported the Shannon Diversity Index in regards the soil having a higher diversity and total taxonomic richness compared to melon samples (S3A–S3D Fig). Principal coordinate analysis demonstrated bacterial composition clustered by the sample type (R$^2$: 0.16; Permutations P-value: 0.001; Permutations 999; Fig 5B). Interestingly, the Central Valley soil samples clustered significantly closer to all the cantaloupe samples compared to the Yuma and Imperial Valley soil samples (Fig 5B).

Taxonomic composition found there were several shared bacterial families between the sample types and the growing region. The top three bacterial families for Yuma Valley for both sample types were *Bacillaceae*, *Chitinophagaceae*, and *Sphingomonadaceae*, those for Imperial Valley were *Bacillaceae*, *Micromonosporaceae*, and *Nocardioidaceae*, and those for the Central Valley were *Bacillaceae*, *Micrococcaceae*, and *Planococcaceae*, which further supports *Bacillaceae* as a major to bacterial family present in commercial cantaloupe fields both in the soil and on the cantaloupe surface (Fig 6A). LEfSe linear discriminant analysis identified overall

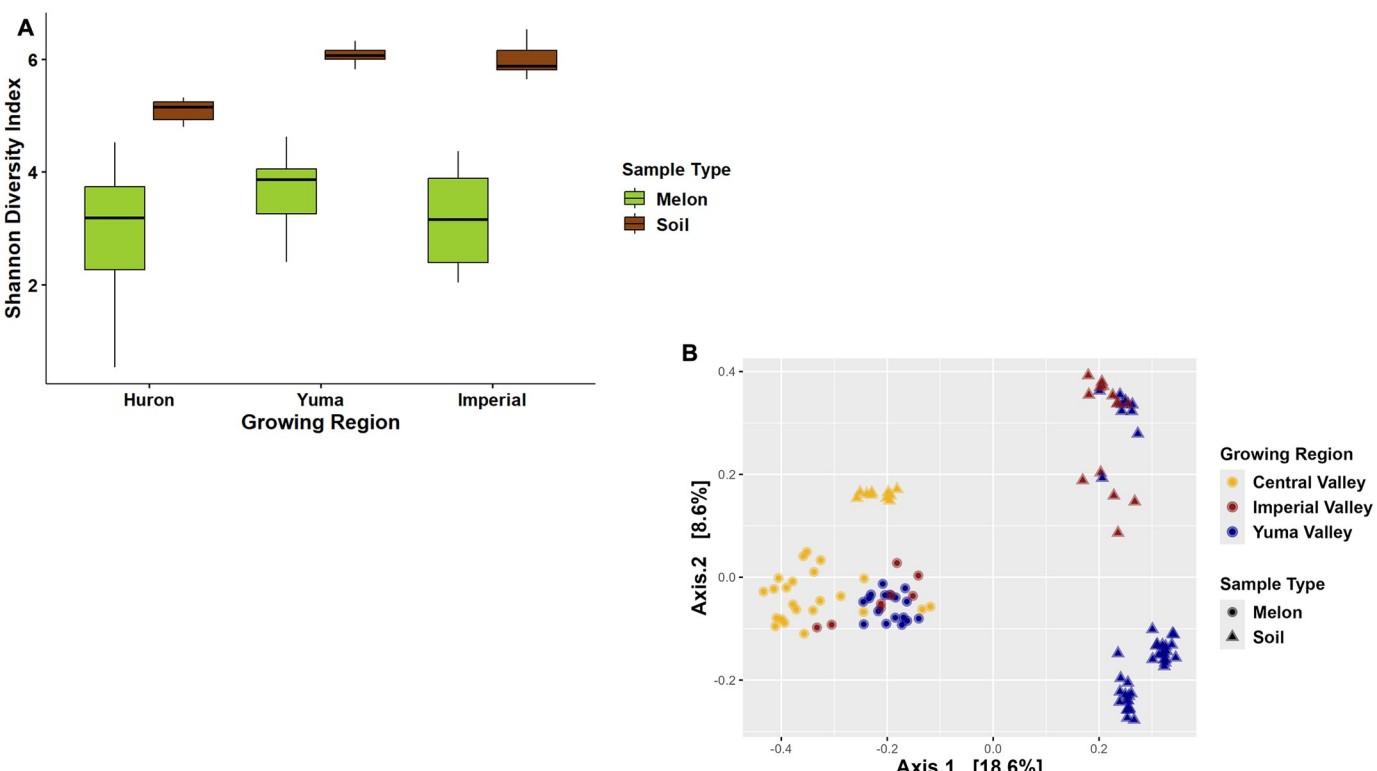

**Fig 5. Bacterial diversity of soil and cantaloupe collected in two commercial agricultural states.** (A) Shannon Diversity Index plotted based on the site for soil and cantaloupe samples. (B) Bray-Curtis PCoA plot clustered by the sample type rather than the regions.

taxonomic features for the three growing regions that were similar to individual soil and cantaloupe samples. There were more features for Central Valley compared to other regions with Imperial Valley having the smallest number of features (Fig 6B). Central Valley had *Enterobacteriaceae* and *Micrococcaceae* as top features that were primarily found in the cantaloupe samples, while other taxonomic features came from the soil samples. Imperial Valley had *Streptomycetaceae* as the top taxonomic feature, with only *Nitrososphaeraceae* found in soil samples individually (Figs 2B & 4B). Yuma Valley had top features like iii1-15 and *Sphingomonadaceae* that were consistent in individual soil and cantaloupe analyses.

Among all the samples including sample type and regions there were two core taxa identified which were *Bacillaceae* and *Micrococcaceae*. Central Valley was the only one to have these taxa in both sample types when sub-sampled to the growing region. Imperial and Yuma Valley had either *Bacillaceae* or *Micrococcaceae* in the melon core, but both were present in soil samples. Therefore, these bacterial families represent the significant soil bacterial families that are potentially transferred to the surface of the cantaloupes in commercial fields. These results indicated that there are regional differences to these key core microbiomes that are shared between the soil and the cantaloupes that are grown in commercial cantaloupe fields.

## Discussion

This is the first study to explore the cantaloupe rind microbiome and soil microbiome from different commercial cantaloupe fields in the United States at the point of harvest, which provides a foundation for cantaloupe producers and the melon industry to understand the

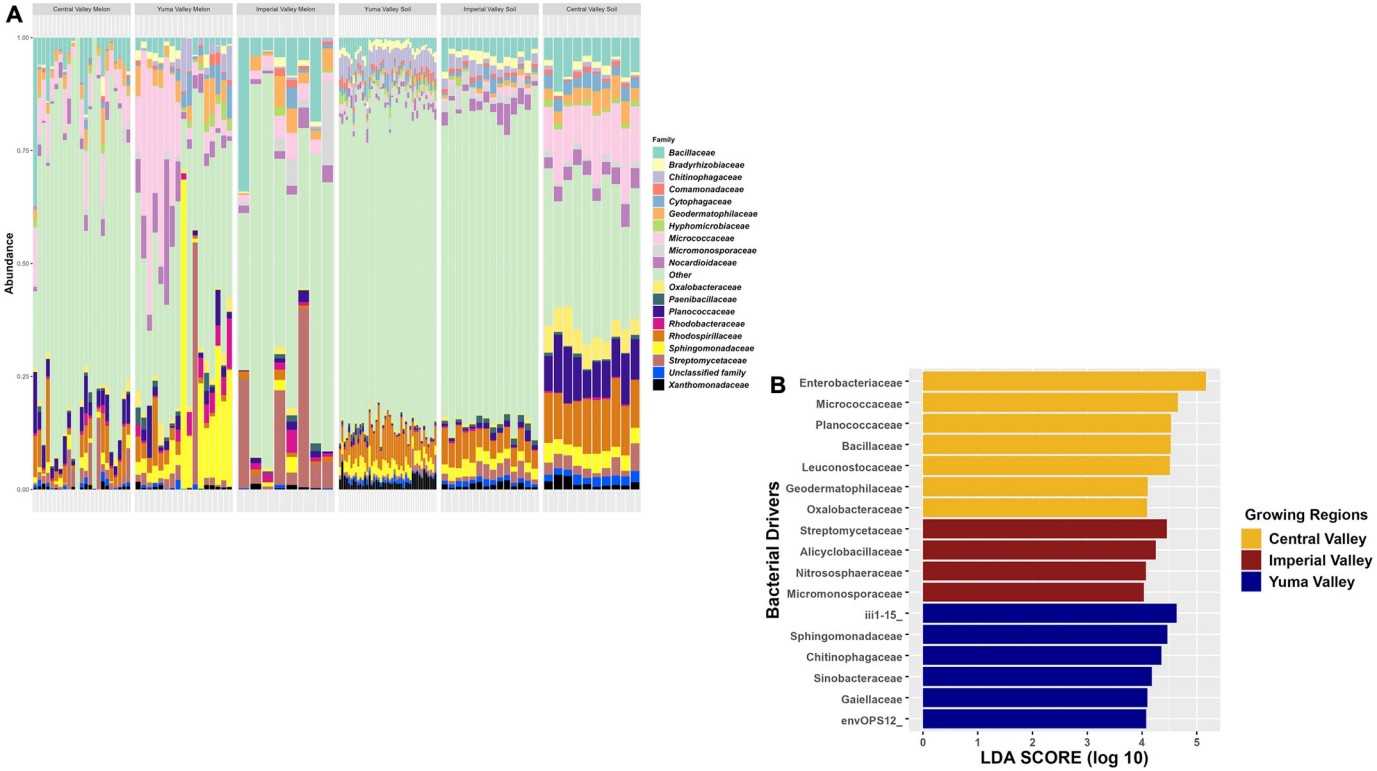

**Fig 6. Bacterial composition of soil and cantaloupe collected in two commercial agricultural states.** (A) Taxonomic relative abundance looking at the regions for soil and cantaloupe samples including the top 18 bacterial families in the samples. The "Other" category is composed of 194 bacterial families, while the "Unclassified family" category represents those reads that could not be taxonomically classified. (B) Lefser analysis for soil and cantaloupe samples looking at sites (LDA = 4).

bacterial communities present on the surface of cantaloupes entering the post-harvest processing stage of production. Additionally, the findings provide information on the variation between three critical growing regions that produce approximately 90% of cantaloupes in the United States [1]. Overall, our study found that bacterial diversity was significantly impacted by the location of the commercial field for both soil and cantaloupe samples. However, the distance between the regions appears to impact the diversity variation, as it was found that two regions, Yuma Valley, Arizona and Imperial Valley, California did not have significant differences for either sample type collected. These regions are approximately 60 miles apart, therefore may be commonly exposed to similar environmental factors during the growing season. Whereas Central Valley, California is over 350 miles away from either of the other two valleys and had significantly different bacterial diversity and taxonomic composition for both soil and melon samples compared to the other two locations.

Regional differences that involved environmental and/or climatic factors have been previously reported to be contributors to bacterial differences in soil [33–36]. From an environmental aspect, certain land usage like agricultural fields with varying types of land usage had varying abundances of bacterial families in the soil, fluctuating among the season that the samples were taken, indicating there can be a temporal influence [33, 37–39]. Climatic factors like relative humidity, annual rain and temperature variations [40] have been shown to reduce certain bacterial families while others thrived [41, 42].

This study also found samples grouped more by sample type, by soil or melon, rather than region, but within sample types there was still clustering by the region. Franco-Frias et al found similar results in that hand rinsate and melon rind samples clustered together but separate from soil samples, indicating differences in soil bacterial composition compared to hand rinsates and rind [10]. Their study also found a temporal difference in the soil samples, as soil samples clustered by the visit period of June and July rather than region, similar to our study that also found temporal clustering from Yuma Valley soil samples. Both of these studies indicate that there are temporal shifts in the bacterial communities during the growing seasons and/or between seasons. However, further experiments are needed within a single region to specifically confirm these temporal shifts in bacteria during and between cantaloupe growing seasons including assessing weather and/or land application impacts on the communities.

Bacterial communities found in the soil can impact the communities found on surfaces of fresh produce, lending to their role in colonization of the surfaces, metabolite bioavailability of the produce in connection to plant health, and their larger role in crop yield and consumer health [33, 43, 44]. Compromising plant health, plant-pathogenic or spoilage microbes can transfer from soil to melon surface through mechanical factors like direct contact between melon surface and the soil, plastic covers when transferred from greenhouses, or directly sown in the ground [6, 20, 45] or through climatic conditions like rain, humidity, or wind transference of the microbes or the dust carrying them from melon to melon. Some of the microbes can be introduced for the first time within a region or be dormant until favorable conditions become available such as the right temperatures or monocropping of the fields [46].

The bacterial families seen in soil alone among the different regions were *Bacillaceae*, *Chitinophagaceae*, and *Gaiellaceae*, whereas Central Valley also had *Micrococcaceae* and *Planococcaceae*. These microbes have a role in soil and plant health such as sustaining different soil processes, surviving during extreme environmental condition, and improving substrate bioavailability [47]. *Bacillus*, a genus in the family *Bacillaceae*, is a common soil microorganism and is ubiquitous to all environments. Previous studies have shown that species in the *Bacillaceae* family have roles in bioremediation of soils as well as biological controls of plant diseases like *Fusarium* wilt as well as play a role in decreasing growth of *Listeria monocytogenes* [48–51]. While *Bacillaceae* was one of the most common bacterial families found in this study, since classification was at the bacterial family level and not genus level, we can only speculate about if *Bacillus* is one of the dominate genera present that could prevent *L. monocytogenes* growth. Although this study does suggest that additional studies examining this subject are worth investigating due to the high levels of *Bacillaceae* on the melon surface. *Chitinophagaceae* members have also been associated with reduction of plant-pathogens like *Fusarium* in alkaline soils, where the higher the pH, the better the ability of this family to reduce the plant disease [52, 53]. *Chitinophagaceae* has also been linked to high phosphorus availability, and processes of cellulose degradation, as well as nitrogen mineralization [54–56]. *Gaiellaceae* has similar roles in sustaining inorganic bioavailability in association with carbon-to-nitrogen ratios [57–59], as well as its abundance in mineral-dense soils [60]. Interestingly, this bacterial family was relatively abundant in Imperial and Yuma Valley soils, where those regions are known to have high alkaline soils with mineral-heavy content [61, 62]. However, Central Valley soils are known to have a more acidic pH as well as higher organic matter, indicating the presence of *Micrococcaceae* due to its abundance in acidic soils [63].

In conjunction to soil, melons had *Bacillaceae* as a top taxonomic family that can play a role in plant-pathogen suppression by sequestering organic matter [49]. *Enterobacteriaceae* and Streptomycetaceae have been seen to play a role in colonization and indication of nitrogen content [64], while *Streptomycetaceae* has also been characterized in previous studies to harbor secondary metabolites that act against plant pathogens like powdery mildew [65, 66].

Furthermore, these bacterial families are consistent with taxonomic families in soil, indicating a role for the soil impacting the cantaloupe microbiome that could play a role in plant health, with the exception of *Enterobacteriaceae*.

An interesting note to make was the absence or below detection limits of *Enterobacteriaceae* in soil samples for all three growing regions; however, it was present on the surfaces of melon samples collected from all growing regions. As mentioned above, post-harvest processing accounts for indicator microorganisms, and some *Enterobacteriaceae* members are considered because of the niche on foods that provide a substantial food source like sugars that make it uncommon to find in soils in high abundances [18, 67, 68]. The presence of *Enterobacteriaceae* on the surfaces of melons and not in the soil could be attributable to the possibility of different organic and inorganic fertilizers being added to the soils when the melons are immature that play a role in how this family colonizes melons [69]. Earlier investigators had seen relative abundance levels decrease in the presence of high nitrogen levels in the soil for *Enterobacteriaceae* [64, 69, 70]. Nitrogen supplementation can be either through a spray or drip system which could impact the soil community more than potentially the surface of the melon. The drip system is most commonly used for melon production, so this system could indicate the low nitrogen content on the melon surface where the *Enterobacteriaceae* family could have been introduced through fertilizers like manure and thrived due to low nitrogen content [20, 71]. However, further studies looking at metabolite and RNA-based approaches for active metabolites produced by the microbes that interact with these inorganic compounds should be assessed with activity of microorganisms in the presence of inorganic compounds on surfaces of produce to substantiate this inference that can play a role in consumer safety and post-harvest processing.

Besides a plant health perspective, there is a consumer perspective as well for soil to melon transference of bacteria or other microorganisms. Similar environmental conditions such as dust [40], rain [40, 41], and humidity [72] play a role in colonization of microorganisms at point of harvest as well as post-harvest at a processing facility from other studies. Other factors like the soil drainage [6, 20], concentration of compounds like nitrogen and phosphorus [37, 73], and pH [10] have effects on microbial colonization. Attachment and colonization of bacteria could make post-harvest processing more difficult in washing the surfaces and passing critical control points acceptable for melons to be released to the consumer [67, 74]. Several studies have observed decreases in attachment of pathogenic microorganisms to fresh produce with different washes and temperature conditions but didn't fully eliminate the microorganism, either human-pathogenic [75–78] or plant-pathogenic [79–81]. In this study, the top bacterial features across the three growing regions found on the cantaloupe surfaces at harvest were *Bacillaceae*, *Enterobacteriaceae*, and *Streptomycetaceae*. Growing cantaloupes in commercial fields means that it is nearly impossible to eliminate the factor of soil to melon contact. This contact leads to the potential of foodborne pathogen colonization or transfer from soil therefore, offering a critical control point of contamination in fields. The study supports that there is potentially transfer of soil bacteria to the melon surface with the presence of either *Bacillaceae* and *Micrococcaceae* in the melon and soil core microbiomes. Although, additional studies should be incorporated to directly see how bacterial communities are transferred from soil to melon due to direct contact in the field to better understand the mechanisms, but this study lays the foundation that there appears to be direct transfer of some bacterial families between soil and melon during the growing period. *Enterobacteriaceae* members are often considered a sign of fecal contamination depending on the genera and species present, and this family also contains several foodborne pathogens including *Salmonella* and Shiga toxin-producing *Escherichia coli* (STEC). *Enterobacteriaceae* members were found on the cantaloupe surface but not in the soil (at least at detectable levels), suggesting that the cantaloupes could

have been exposed to fecal contamination in the commercial fields from another source, such as a certain management practice, or that even small amounts of *Enterobacteriaceae* in the soil could potentially result in colonization such as certain pathogens. Therefore, understanding the bacterial communities present on the melons and in the soil from these fields can help in further understanding pathogen colonization and development of potential interventions to increase melon safety.

We determined only two core bacterial families for both soil and melon samples collected from all of the regions that were *Bacillaceae* and *Micrococcaceae*, where these two members have roles in both soil and plant health. The *Bacillaceae* family included only *Bacillus flexus*, which has been used in biodegradation of toxins and plastics, as well as a biocontrol agent against different plant diseases that include *Fusarium* wilt [48, 50]. The *Micrococcaceae* family included *Arthrobacter*, which studies have found in higher relative abundance associated with hotter weather conditions and drought-tolerant shrubs [82], which is interesting, as melons, particularly cantaloupes, are grown in well-drained soil with low water retention [2, 6, 20], which could explain their presence. Overall, the core microbiome identified from the soil and cantaloupe samples of the three regions all have similar phenotypes in common such as anaerobic activity, drought-tolerance, and survival under harsh conditions. Yuma and Imperial Valley are relatively closer in distance and share similar characteristics of dry heat, while Central Valley has more of a temperate climate, with differences in temperature ranging from -13 to -12˚C during the summer months [83], with relatively more humid days during [83] growing seasons. Understanding the differences among bacteria that are detected on the surfaces of melons at harvest lay the foundation to effectively design appropriate studies to help develop optimum post-harvest processing methods and storage conditions to maximize melon safety and quality before melons arrive at the retail stores for consumer purchase.

## Conclusion

In this study, the differences in bacterial diversity and composition between the cantaloupe and soil microbiomes from commercial cantaloupe fields in three major growing regions in the US were explored. The results of this study highlighted that soil samples collected from all three growing regions were more diverse as well as clustered by the growing region. Cantaloupe samples had less variation in bacterial diversity among the growing regions, however, these samples did have *Enterobacteriaceae* present that was not detected in any of the soil samples. Lastly, there were two core families *(Bacillaceae* and *Micrococcaceae)* that were present in both sample types across all three regions. These families exhibit drought-tolerance and are durable in environments of high stress. This study provides a foundation of information about the bacterial communities present in commercial cantaloupe fields in major growing regions in the US. Such information allows for further characterizing, to help the melon industry understand bacterial interactions upon entering post-harvest processing and storage, thereby, contributing towards devising methods for improving melon safety and quality.

## Supporting information

**S1 Fig. Bacterial diversity compared across several alpha matrices for soil across two commercial agricultural fields.** (A) Shannon Diversity Index plotted based on the site for the soil samples considering both richness and evenness. (B) Chao1 total richness plotted based on the site for the soil samples considering total taxonomic ASV richness. (C) Simpson Evenness plotted based on the site for the soil samples considering distribution (evenness) of the bacteria across the samples. (D) Simpson Dominance Index plotted based on the site for the soil

samples considering inverse richness.
(TIF)

**S2 Fig. Bacterial diversity compared across several alpha matrices for melons across two commercial agricultural fields.** (A) Shannon Diversity Index plotted based on the site for the melon samples considering both richness and evenness. (B) Chao1 total richness plotted based on the site for the melon samples considering total taxonomic ASV richness. (C) Simpson Evenness plotted based on the site for the melon samples considering distribution (evenness) of the bacteria across the samples. (D) Simpson Dominance Index plotted based on the site for the melon samples considering inverse richness.
(TIF)

**S3 Fig. Bacterial diversity compared across several alpha matrices for soil and melons across two commercial agricultural fields.** (A) Shannon Diversity Index plotted based on the site for the soil and melon samples considering both richness and evenness. (B) Chao1 total richness plotted based on the site for the soil and melon samples considering total taxonomic ASV richness. (C) Simpson Evenness plotted based on the site for the soil and melon samples considering distribution (evenness) of the bacteria across the samples. (D) Simpson Dominance Index plotted based on the site for the soil and melon samples considering inverse richness.
(TIF)

**S1 File. Supplemental statistics.** Supplemental information providing all the statistical tests used, parameters used for each statistical test, and the resulting p-values for each test that was run on all the data used in the study.
(DOCX)

**S2 File. Supplemental tables.** Includes a total of nine supplemental tables including: S1 Table. Reads per sample type across rarefaction limits; S2 Table. Core taxa shared among all regions for soil samples; S3 Table. Core taxa found in Yuma Valley for soil samples; S4 Table. Core taxa found in Imperial Valley for soil samples; S5 Table. Core taxa found in Central Valley for soil samples; S6 Table. Core taxon in all regions for melon samples; S7 Table. Core taxa found in Central Valley for melon samples; S8 Table. Core taxa found in Imperial Valley for melon samples; S9 Table. Core taxa found in Yuma Valley for melon samples.
(DOCX)

## Acknowledgments

The authors thank our melon industry collaborators who generously provided access to their fields for harvesting melon and soil samples for the study. Additionally, we thank all the members of the Cooper laboratory for providing critical technical assistance in sample processing and other aspects of the study.

## Author Contributions

**Conceptualization:** Madison Goforth, Kerry K. Cooper.

**Formal analysis:** Madison Goforth.

**Funding acquisition:** Paul Brierley, Bhimanagouda Patil, Sadhana Ravishankar.

**Investigation:** Madison Goforth, Victoria Obergh, Steven Huynh, Craig T. Parker.

**Methodology:** Madison Goforth, Victoria Obergh, Richard Park, Martin Porchas, Tom Turni, Steven Huynh, Craig T. Parker, Kerry K. Cooper.

**Project administration:** Kerry K. Cooper.

**Resources:** Kerry K. Cooper.

**Supervision:** Kerry K. Cooper.

**Writing – original draft:** Madison Goforth, Kerry K. Cooper.

**Writing – review & editing:** Madison Goforth, Victoria Obergh, Richard Park, Martin Porchas, Paul Brierley, Tom Turni, Bhimanagouda Patil, Sadhana Ravishankar, Steven Huynh, Craig T. Parker, Kerry K. Cooper.

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
