## [Decision Letter · Decision Letter 0]

7 Mar 2024

PONE-D-23-43395Bacterial diversity of cantaloupes and soil from Arizona and California commercial fields at the point of harvestPLOS ONE

Dear Dr. Cooper,

Thank you for submitting your manuscript to PLOS ONE. After careful consideration, we feel that it has merit but does not fully meet PLOS ONE’s publication criteria as it currently stands. Therefore, we invite you to submit a revised version of the manuscript that addresses the points raised during the review process.

Please submit your revised manuscript by Apr 21 2024 11:59PM. If you will need more time than this to complete your revisions, please reply to this message or contact the journal office at plosone@plos.org. Please include the following items when submitting your revised manuscript:A rebuttal letter that responds to each point raised by the academic editor and reviewer(s). You should upload this letter as a separate file labeled 'Response to Reviewers'.A marked-up copy of your manuscript that highlights changes made to the original version. You should upload this as a separate file labeled 'Revised Manuscript with Track Changes'.An unmarked version of your revised paper without tracked changes. You should upload this as a separate file labeled 'Manuscript'.

We look forward to receiving your revised manuscript.

Kind regards,

Ying Ma, Ph.D.

Academic Editor

PLOS ONE

Journal Requirements:

This study was supported by the USDA-NIFA- SCRI # 2017-51181-26834 through the National Center of Excellence for Melon at the Vegetable and Fruit Improvement Center of Texas A&M University, and Technology and Research Initiative Fund (TRIF) provided to Kerry Cooper by the University of Arizona. No funding agency had any role in the study design, data collection and analysis, decision to publish, or preparation of the manuscript.

Please include your amended Funding Statement within your cover letter. We will change the online submission form on your behalf

4. Please note that your Data Availability Statement is currently missing [the repository name and/ the DOI/ a direct link to access each database]. If your manuscript is accepted for publication, you will be asked to provide these details on a very short timeline. We therefore suggest that you provide this information now, though we will not hold up the peer review process if you are unable.

Reviewers' comments:

Reviewer's Responses to Questions

**Comments to the Author**

1. Is the manuscript technically sound, and do the data support the conclusions?

Reviewer #1: Partly

Reviewer #2: Yes

2. Has the statistical analysis been performed appropriately and rigorously? 

Reviewer #1: Yes

Reviewer #2: Yes

3. Have the authors made all data underlying the findings in their manuscript fully available?

Reviewer #1: Yes

Reviewer #2: Yes

4. Is the manuscript presented in an intelligible fashion and written in standard English?

Reviewer #1: Yes

Reviewer #2: Yes

5. Review Comments to the Author

Reviewer #1: General Comments

This manuscript describes the characterization of the microbiome from the surface of cantaloupe rinds and soil from cantaloupe growing regions. Microbial communities were compared at the Family taxonomic rank. While interesting, it was largely a survey of the microbial community. A stronger case needs to be made in the introduction and discussion regarding the relevancy of this work for food quality and safety.

There are numerous small typographical errors (e.g., no space between a word and reference. For example, on line 92: “December[7].” Instead of “December [7].”).

The introduction (specifically the final 2 paragraphs) focus on pathogen contamination. However, the overall project goals (Lines 143 – 145) don’t explicitly address how this work can potentially contribute to pathogen mitigation.

Rarefying to 1,500 reads (Line 214) is concerning. A typical MiSeq run can easily generate 20 – 50,000 reads per sample so this suggests at least some low-quality samples. What were the range and average reads per sample before and after quality filtering? The highly variable bacterial diversity (Figures 4A and 6A) and high number of taxa classified as “other” suggests poor read quality. What percentage of reads were classified at each taxonomic rank?

In the discussion, several specific genera such as Bacillus and Listeria were mentioned along with functional roles of several other taxa. In the results section, all the comparisons were performed at a family level, so it is speculative to compare the roles of specific genera and their potential function. This should be clarified.

Specific comments

Lines 88 – 93: This doesn’t really contribute anything of importance to the subject of the manuscripts so you may want to consider removing.

The second paragraph (lines 94 – 102) should be moved to the beginning of the introduction since it provides a better introduction to the topic of the paper.

Lines 126 – 130: If pathogen contamination is a post-harvest risk, how does knowledge of the microbiome preharvest contribute to risk reduction? This is a tenuous connection that needs to be better justified.

Line 129: Replace “sense” with “since”

Line 153: Were the cantaloupe varieties the same at all locations? If not, could this contribute some of the differences that were observed in rind communities from different locations?

Lines 160 – 164: Methods are not clear here. Was the surface sterilized prior to collecting a sample or was the swab used to collect one sample by swabbing the entire surface not in contact with the ground?

Line 216: What was the rationale for using Shannon evenness index?

Lines 371 – 373: This sentence appears incomplete.

Line 398: Replace “From the” with “From an”

Line 455: Clarify that Enterobacteriaceae was below detection levels not necessarily absent.

Line 490 – 491: The potential for ground contact to lead to pathogen colonization is a reasonable claim. How do the results of this study provide support for this statement?

Line 495 – 497: What is known about the management practices in these fields? Were there practices that could have led to fecal contamination? Is it possible that Enterobacteriaceae was present in the soil below detection limits?

Lines 521 – 525: Can this be concluded from the results? Wouldn’t additional sampling be needed after post-harvest processing to determine how the community changes during post-harvest processing?

Line 536: Replace “sturdy” with “study”

References are not formatted consistently. Some contain DOI, PMID, and PMCID. Are all three required by the journal? Reference 29 appears incomplete.

Reviewer #2: I strongly suggest changing the "Shannon evenness index" to "Shannon index" or "Shannon richness index" all over the manuscript. The Simpson index determines evenness.

Also, after referring to data from the Shannon index inference, change "diversity" for "richness" throughout the manuscript.

In the Figures, also change "Shannon Diversity" for "Shannon Index".

6. PLOS authors have the option to publish the peer review history of their article (what does this mean?). If published, this will include your full peer review and any attached files.

Reviewer #1: No

Reviewer #2: **Yes: **Aarón Barraza

---

## [Author Response · Author response to Decision Letter 0]

16 Jun 2024

Response to reviewers:

We thank the reviewers for their comments and excellent insights into improving our manuscript prior to publication. We have addressed all the comments in the revised manuscript and the changes for each specific comment are highlighted below.

Reviewer #1: General Comments

This manuscript describes the characterization of the microbiome from the surface of cantaloupe rinds and soil from cantaloupe growing regions. Microbial communities were compared at the Family taxonomic rank. While interesting, it was largely a survey of the microbial community. A stronger case needs to be made in the introduction and discussion regarding the relevancy of this work for food quality and safety.

There are numerous small typographical errors (e.g., no space between a word and reference. For example, on line 92: “December[7].” Instead of “December [7].”).

Comment is noted and implemented throughout the manuscript.

The introduction (specifically the final 2 paragraphs) focus on pathogen contamination. However, the overall project goals (Lines 143 – 145) don’t explicitly address how this work can potentially contribute to pathogen mitigation.

We have further clarified in the manuscript that knowing the variation of critical control points in the HACCP protocol for melons increases risk of pathogen contamination, so understanding the surface bacteria and the community could help standardize this procedure by knowing what is on the surface. Additionally, we have added/clarified that to begin to study and understand pathogen interactions with melon microbiome on the surface as a first step in pathogen colonization/contamination requires we understand who is there as part of the melon microbiome and how those communities change in different growing regions. Furthermore, this also goes to the soil microbiome in the fields as this is a potential source of certain members of the melon microbiome prior to harvest. This leaves a critical food safety knowledge gap about pathogen-microbe interactions that can occur during the post-harvest processing when contamination risks are higher.

Rarefying to 1,500 reads (Line 214) is concerning. A typical MiSeq run can easily generate 20 – 50,000 reads per sample so this suggests at least some low-quality samples. What were the range and average reads per sample before and after quality filtering? The highly variable bacterial diversity (Figures 4A and 6A) and high number of taxa classified as “other” suggests poor read quality. What percentage of reads were classified at each taxonomic rank?

Overall, there was a low biomass of melon surfaces, as we only sampled a single cantaloupe with a single swab to represent a sample instead of doing composite samples as has been done in other studies to get higher biomass, which resulted in a need for rarefaction of these samples to 1,500. We originally, did the rarefication for all melon and soil samples to 1,500 to keep the samples equal. However, in the revised manuscript we have redone the analysis where we have rarefied for each sample type specifically, which resulted in rarefication at 20,000 sample depth for soil, and kept the 1,500 reads for melons as it would lose too many samples if it increased to 2,000. For overall direct comparative analysis between soil and melon samples, rarefaction stayed at 1,500 to avoid sequencing depth bias of bacterial communities. We have added a Supplemental Table S1 that includes the range of reads for the different sample types, average reads per sample before filtering, average reads per sample after quality filtering, rarefaction levels, number of taxa prior to rarefying, and number of taxa after rarefying. We understand the confusion with the “Other” category in Figures 2A, 4A, and 6A, but these are not unclassified reads just bacterial families that were not in the top 51 (Figure 2A), top 11 (Figure 4A), or top 18 (Figure 6A), and more an indication of the bacterial diversity of the samples as opposed to poor read quality. For example, in Figure 6A there is an “Unclassified family” group that is unclassified reads that is very small (<1% - 2% of the sample reads), and Figure 4A does not have an “Unclassified family” category because there is such a small fraction of the reads that were unclassified. We have hopefully made it a little more clear in the Figure legends for Figure 2, Figure 4, and Figure 6. 

In the discussion, several specific genera such as Bacillus and Listeria were mentioned along with functional roles of several other taxa. In the results section, all the comparisons were performed at a family level, so it is speculative to compare the roles of specific genera and their potential function. This should be clarified.

We appreciate the comment, and we have clarified in the discussion those sections referring to specific genera that this study only classified to the family level, therefore we are only speculating that Bacillus is a dominate genera in the Bacillaceae identifying in this study. 

Specific comments

Lines 88 – 93: This doesn’t really contribute anything of importance to the subject of the manuscripts so you may want to consider removing.

We appreciate the guidance to improving the manuscript and have addressed the comment by explaining further and hopefully clarifying that knowing temporal geography will help characterize surface bacterial communities particularly those with less variation in regional growth. 

The second paragraph (lines 94 – 102) should be moved to the beginning of the introduction since it provides a better introduction to the topic of the paper.

Again, we thank the reviewer for the comment allowing us to improve the manuscript, the comment has been noted and implemented in the manuscript.

Lines 126 – 130: If pathogen contamination is a post-harvest risk, how does knowledge of the microbiome preharvest contribute to risk reduction? This is a tenuous connection that needs to be better justified.

We thank the reviewer for the comment, and we have implemented changes in the manuscript to clarify the connection, particular that understanding what is on the surface of the melon at the point of harvest allows for a better understanding of what is entering the post-harvest processing stage. Additionally, in the next paragraph we further explain and also discuss why critical control points reduce potential pathogenic bacterial contamination, however, understanding the surface bacterial communities could help standardize protocols and reduce points of contamination risks. 

Line 129: Replace “sense” with “since”

Comment is noted and implemented in the manuscript

Line 153: Were the cantaloupe varieties the same at all locations? If not, could this contribute some of the differences that were observed in rind communities from different locations?

Comment is noted and the reviewer brings up an interesting point. However, we do not believe that the varieties had a significant impact in this study, as cantaloupe varieties were the same between Central and Imperial valley. Statistical analysis showed no difference among Yuma Valley and the other two valleys in diversity, indicating little variation and that variety of the melon was not dependent on the outcome of bacterial diversity at for this study.

Lines 160 – 164: Methods are not clear here. Was the surface sterilized prior to collecting a sample or was the swab used to collect one sample by swabbing the entire surface not in contact with the ground?

We appreciate the reviewer helping us to clarify methods, the comment is noted and manuscript changes done for clarity.

Line 216: What was the rationale for using Shannon evenness index?

The rationale for Shannon evenness index (now being changed to Shannon Diversity index per other reviewers recommendation) was to determine both richness and evenness of bacterial communities at the alpha-diversity level, so between samples. Since Shannon Diversity index takes into account both richness and evenness, it is a parameter that is highly utilized in other microbiome studies relating to fresh produce. While you cannot directly compare Shannon Diversity index between studies, we still wanted to keep consistent analysis from other studies for context. We have also included other alpha matrices like Simpson evenness, Simpson dominance, and Chao in the Supplemental figures. 

Lines 371 – 373: This sentence appears incomplete.

Again, we thank the reviewer for helping improve the manuscript, we have re-written the sentence. 

Line 398: Replace “From the” with “From an”

Comment is noted and implemented in the manuscript

Line 455: Clarify that Enterobacteriaceae was below detection levels not necessarily absent.

Comment is noted and implemented in the manuscript

Line 490 – 491: The potential for ground contact to lead to pathogen colonization is a reasonable claim. How do the results of this study provide support for this statement?

We thank the reviewer for helping us to clarify and support our statements better, we have made changes to the manuscript to support the statement. Particularly, we have emphasized that the claim is backed up with the results of inferring what Family-level bacteria are found in soil and on melons that are present on either both, soil, or just melons. 

Line 495 – 497: What is known about the management practices in these fields? Were there practices that could have led to fecal contamination? Is it possible that Enterobacteriaceae was present in the soil below detection limits?

The reviewer makes an excellent point, unfortunately management practices in these fields were not noted for this study. We have added changes to the manuscript to clarify this point and also mention that management practices might have had a role and/or soil levels of Enterobacteriaceae could be below detection limits. Further research that would account for practices and further approaches would give us better insight to understand the role certain management practices might have in fecal contamination of the melons. 

Lines 521 – 525: Can this be concluded from the results? Wouldn’t additional sampling be needed after post-harvest processing to determine how the community changes during post-harvest processing?

We thank the reviewer for pointing this out, and we have re-written the sentence to emphasize how the results of this study lay the foundation, but additional studies are needed to effective develop post-harvest processing to improve melon safety. 

Line 536: Replace “sturdy” with “study”

Comment is noted, but sturdy is the correct wording to describe the bacterial family members exhibiting “sturdy” characteristics investigated from other studies. We have changed “sturdy” to “durable” to help clarify any confusion.

References are not formatted consistently. Some contain DOI, PMID, and PMCID. Are all three required by the journal? Reference 29 appears incomplete.

Comment is noted and implemented in the manuscript for reference 29. 

PloS bibliographic format will incorporate all three if included in the citation. 

Reviewer #2: I strongly suggest changing the "Shannon evenness index" to "Shannon index" or "Shannon richness index" all over the manuscript. The Simpson index determines evenness. Also, after referring to data from the Shannon index inference, change "diversity" for "richness" throughout the manuscript.

We thank reviewer 2 for the guidance and suggests for improving the manuscript and making things clearer. We have changed “Shannon evenness index” to “Shannon Diversity Index” as the index accounts for both richness and evenness. To be consistent to the accurate measurement, we agree that the naming convention needed to be changed; but we respectively disagree that it should be “richness,” since the measurement accounts for evenness as well. 

In the Figures, also change "Shannon Diversity" for "Shannon Index".

Comment is noted and changed to “Shannon Diversity Index” as it is an index of richness and evenness of the samples

---

## [Decision Letter · Decision Letter 1]

8 Jul 2024

Bacterial diversity of cantaloupes and soil from Arizona and California commercial fields at the point of harvest

PONE-D-23-43395R1

Dear Dr. Cooper,

We’re pleased to inform you that your manuscript has been judged scientifically suitable for publication and will be formally accepted for publication once it meets all outstanding technical requirements.

Kind regards,

Ying Ma, Ph.D.

Academic Editor

PLOS ONE

Additional Editor Comments (optional):

Reviewers' comments:

Reviewer's Responses to Questions

**Comments to the Author**

1. If the authors have adequately addressed your comments raised in a previous round of review and you feel that this manuscript is now acceptable for publication, you may indicate that here to bypass the “Comments to the Author” section, enter your conflict of interest statement in the “Confidential to Editor” section, and submit your "Accept" recommendation.

Reviewer #1: (No Response)

Reviewer #2: All comments have been addressed

2. Is the manuscript technically sound, and do the data support the conclusions?

Reviewer #1: Partly

Reviewer #2: Yes

3. Has the statistical analysis been performed appropriately and rigorously? 

Reviewer #1: Yes

Reviewer #2: Yes

4. Have the authors made all data underlying the findings in their manuscript fully available?

Reviewer #1: Yes

Reviewer #2: Yes

5. Is the manuscript presented in an intelligible fashion and written in standard English?

Reviewer #1: Yes

Reviewer #2: Yes

6. Review Comments to the Author

Reviewer #1: (No Response)

Reviewer #2: (No Response)

7. PLOS authors have the option to publish the peer review history of their article (what does this mean?). If published, this will include your full peer review and any attached files.

Reviewer #1: No

Reviewer #2: **Yes: **Aarón Barraza

---

## [Editor Report · Acceptance letter]

8 Aug 2024

PONE-D-23-43395R1 

PLOS ONE

Dear Dr. Cooper, 

I'm pleased to inform you that your manuscript has been deemed suitable for publication in PLOS ONE. Congratulations! Your manuscript is now being handed over to our production team.

Kind regards, 

on behalf of

Dr. Ying Ma 

Academic Editor

PLOS ONE